# Dissolved-$Cl_2$ triggered redox reaction enables high-performance perovskite solar cells

Yujie Luo[1,2], Kaikai Liu[1,2], Liu Yang[1], Wenjing Feng[1], Lingfang Zheng[1], Lina Shen[1], Yongbin Jin[1], Zheng Fang[1], Peiquan Song[1], Wanjia Tian[1], Peng Xu[1], Yuqing Li[1], Chengbo Tian[1], Liqiang Xie ®[1] ✉ & Zhanhua Wei ®[1] ✉

Constructing 2D/3D perovskite heterojunctions is effective for the surface passivation of perovskite solar cells (PSCs). However, previous reports that studying perovskite post-treatment only physically deposits 2D perovskite on the 3D perovskite, and the bulk 3D perovskite remains defective. Herein, we propose $Cl_2$-dissolved chloroform as a multifunctional solvent for concurrently constructing 2D/3D perovskite heterojunction and inducing the secondary growth of the bulk grains. The mechanism of how $Cl_2$ affects the performance of PSCs is clarified. Specifically, the dissolved $Cl_2$ reacts with the 3D perovskite, leading to Cl/I ionic exchange and Ostwald ripening of the bulk grains. The generated $Cl^-$ further diffuses to passivate the bulk crystal and buried interface of PSCs. Hexylammonium bromide dissolved in the solvent reacts with the residual $PbI_2$ to form 2D/3D heterojunctions on the surface. As a result, we achieved high-performance PSCs with a champion efficiency of 24.21% and substantially improved thermal, ambient, and operational stability.

Organic-inorganic hybrid perovskite solar cells have attracted extensive attention due to their high power conversion efficiency (PCE), low cost, and facile processing[1–3]. To date, the highest certified PCE of perovskite solar cells (PSCs) has achieved 25.8%[4]. High-quality perovskite film is the key factor in achieving high PCE. Researchers have made great efforts to regulate perovskite growth through compositional engineering[5–7], processing optimization[8–12], solvent engineering[13–18], and additive engineering[19–22]. In general, it is inevitable to generate defects during perovskite film growth, and surface passivation is an important and effective strategy to reduce the defects. In the past, various passivation agents like Lewis acids and bases[23], halide salts[24], ionic liquids[25], and other organic molecules[26] have been used to passivate the perovskite films. Among them, constructing 2D/3D perovskite heterostructures using alkyl ammonium halide possesses many advantages: (1) The defective 3D perovskite surface can be converted to high-quality 2D perovskite with reduced surface defects; (2) 2D/3D perovskite stacking facilitates the

charge transfer/extraction due to the extra built-in field; (3) The modified interface adjusts the energy level arrangement of perovskite films with the adjacent charge transporting layer; (4) The bulky organic spacer prevents ion migration and water erosion, thus improving the stability of the material[27–30].

The most widely used method for constructing 2D/3D heterojunctions is spin-coating bulky alkyl ammonium halides onto the 3D perovskite, followed by a thermal annealing process. For example, Yoo et al. utilized hexylammonium hydrobromide (HABr) as the 2D perovskite precursor to construct the 2D/3D structure[31]. They found that HABr can be selectively dissolved in chloroform (CF), which was a non-solvent for perovskite films. Therefore, treating the 3D perovskite films with HABr/CF could construct a 2D perovskite layer without damage that was commonly observed in the post-treatment by using alkyl ammonium salt dissolved in isopropanol (IPA). Therefore, this strategy can help fabricate high-performance perovskite solar cells with improved stability. However, the resultant 2D

[1]Xiamen Key Laboratory of Optoelectronic Materials and Advanced Manufacturing, Institute of Luminescent Materials and Information Displays, College of Materials Science and Engineering, Huaqiao University, Xiamen 361021, P.R. China. [2]These authors contributed equally: Yujie Luo, Kaikai Liu.
✉e-mail: lqxie@hqu.edu.cn; weizhanhua@hqu.edu.cn

perovskite capping layer in previous reports was most likely physically stacked onto the 3D perovskite, and only weak interfacial interactions existed between the 2D and 3D perovskite layers[30]. Moreover, this method can't modulate the chemical composition and crystal domains of the underlying 3D perovskite. Therefore, the inside part of the 3D perovskite film is still defective and needs further improvement.

It was reported that introducing chloride compounds can improve the quality of perovskite film in previous studies[24,31–33]. Ye et al. reported that methylammonium chloride (MACl) could adjust the intermediate-related perovskite crystallization and improve the crystal quality[34]. Mahmud et al. demonstrated that octylammonium chloride (OACl) treatment could induce the diffusion of $Cl^-$ into the bulk of the underlying 3D perovskite, which ensures effective passivation[24]. Compared to the chloride compound, the more reactive and aggressive gaseous halogen may chemically and crystallographically reconstruct the bottom 3D perovskite film[32]. Inspired by these reports, we assume that dissolved-$Cl_2$ is promising in renovating the quality of bulk perovskite, which is probably able to achieve chemically soldered 2D/3D perovskite heterojunction for high-performance PSCs.

Herein, we report adopting chlorine-dissolved chloroform ($Cl_2$-CF) as a multifunctional solvent for selectively dissolving HABr to construct 2D/3D perovskite heterojunction, as well as to induce secondary growth of perovskite grains and defect passivation through the redox reaction between $Cl_2$ and $I^-$. During the redox reaction, Cl/I ionic exchange was achieved, and the secondary growth of perovskite grains occurred through the Ostwald ripening, forming a Cl-doped perovskite film with larger grains. Moreover, the incorporated $Cl^-$ could diffuse into the bulk and the buried interface of PSCs, enabling defect passivation inside the device. On the surface of the 3D perovskite, HABr reacted with the defective surface and obtained a high-quality 2D capping layer. The generated Ruddlesden–Popper (RP-type) 2D/3D perovskite heterojunctions achieved both effects of perovskite crystal regrowth and surface passivation. As a result, the champion PSC delivered a high efficiency of 24.21% with negligible hysteresis. The optimized devices showed largely enhanced operational stability, which retained 80% of the initial efficiency after 905 h under continuous one-sun illumination at the maximum power point.

## Results

### Formation of $Cl_2$-CF and its effect on the perovskite film

In this work, the $Cl_2$-CF was obtained by illuminating the fresh chloroform ($CHCl_3$, CF) with a xenon lamp under ~35% relative humidity. The chemical equations in Fig. 1a describe the species' transformation. The chloroform ($CHCl_3$) firstly reacted with oxygen ($O_2$) under light illumination to form phosgene ($COCl_2$) and hydrogen chloride (HCl), and $COCl_2$ further reacted with $O_2$ to form carbon dioxide ($CO_2$) and chlorine ($Cl_2$), leading to the desired $Cl_2$-CF with the strong oxidizing property. We performed chemical analysis experiments to confirm the composition of the

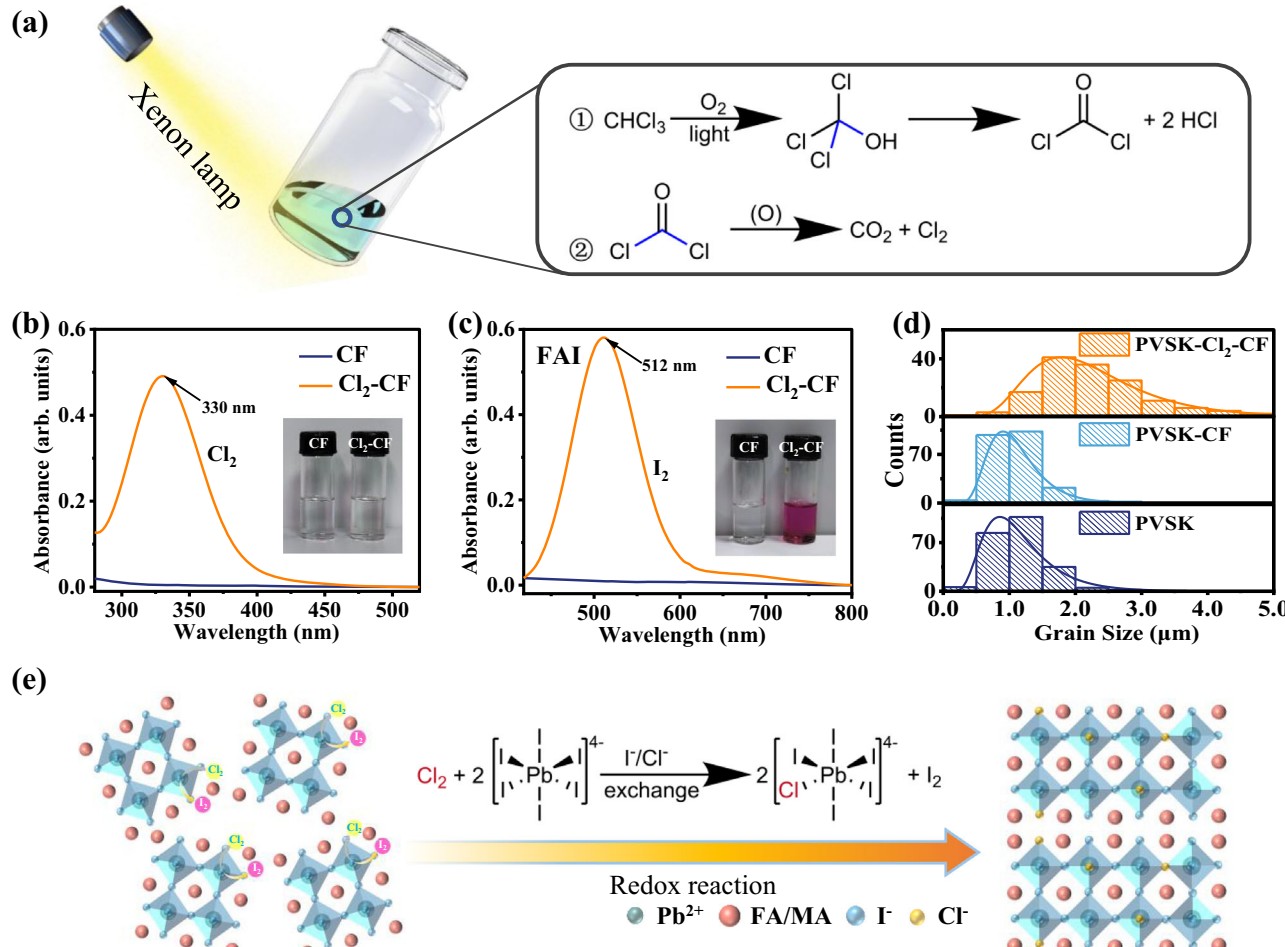

**Fig. 1 | Formation of $Cl_2$-CF and its effect on the perovskite film. a** Diagram of the formation process of the $Cl_2$-CF. UV-vis absorption spectra of **b** CF and $Cl_2$-CF and **c** FAI dissolved in CF and $Cl_2$-CF. **d** Grain size distribution of PVSK, PVSK-CF, and PVSK-$Cl_2$-CF. **e** Schematic illustration of the redox reaction between the $Cl_2$-CF and perovskite film.

$Cl_2$-CF. The barium hydroxide ($Ba(OH)_2$) test in Supplementary Fig. 1 confirmed the presence of $CO_2$ in the $Cl_2$-CF, while the silver nitrate ($AgNO_3$) test in Supplementary Fig. 2 proved the presence of $Cl^-$. We also used the wet starch potassium iodide test paper to check the presence of $Cl_2$ in the $Cl_2$-CF. When the test paper was placed above the $Cl_2$-CF, it immediately changed to blue color as $Cl_2$ could oxidize $I^-$ to get $I_2$ (Supplementary Fig. 3). We further found that the $Cl_2$-CF could make I-containing species turn purple-red and Br-containing ones turn orange-yellow, while Cl-containing species showed no color change (Supplementary Fig. 4). UV-vis absorption spectra in Fig. 1b showed that the $Cl_2$-CF presents a characteristic absorption peak of $Cl_2$ at 330 nm while the CF has no signal[35,36]. These characterizations confirmed the formation of $Cl_2$-CF under light illumination in humid air.

To reveal the reactivity of the $Cl_2$-CF, we conducted a series of UV-vis absorption spectroscopy measurements for each precursor of perovskite dissolved in CF or $Cl_2$-CF. Lead iodide ($PbI_2$), formamidinium iodide (FAI), and methylammonium iodide (MAI) dissolved in $Cl_2$-CF showed UV-vis absorption peaks at 508, 512, and 510 nm[37], respectively (Supplementary Fig. 5a, Fig. 1c, and Supplementary Fig. 5b). Because of the redox reaction between $Cl_2$ and $I^-$, $I_2$ can be formed and lead to a UV-vis absorption peak at 508 nm[37]. Similarly, the oxidative $Cl_2$ in the $Cl_2$-CF can oxidize $Br^-$ ions to obtain $Br_2$. Supplementary Fig. 6a, b showed that methylammonium bromide (MABr) and HABr dissolved in $Cl_2$-CF exhibited UV-vis absorption peaks at 411 and 410 nm[38], respectively. However, for the MACl dissolved in the $Cl_2$-CF, there is no color change except that a UV-vis absorption peak (330 nm) belonging to the $Cl_2$ (Supplementary Fig. 7) was observed[36].

We studied the effect of the solvent (CF and $Cl_2$-CF) on the perovskite film with a composition of $(Cs_{0.05}FA_{0.75}MA_{0.20})$ $Pb(I_{0.96}Br_{0.04})_3$[39] (Supplementary Fig. 8a–c). The control perovskite, CF-treated perovskite, and $Cl_2$-CF treated perovskite are denoted as PVSK, PVSK-CF, and PVSK-$Cl_2$-CF, respectively. The morphology of PVSK-CF was similar to that of the PVSK, where abundant $PbI_2$ (white particles) spread on the surface of perovskite films. In contrast, the PVSK-$Cl_2$-CF presented enlarged perovskite grains with small white particles appearing at the grain boundaries. We ascribed this morphology change to the redox reaction induced by $Cl_2$, and it triggered the secondary growth of the perovskite grains via Ostwald ripening, increasing the crystal size to as large as 5 μm (Fig. 1d). Meanwhile, as shown in Supplementary Fig. 8d–f and g-i, the surface of PVSK-$Cl_2$-CF showed more smooth morphology with a roughness of 28.2 nm (32.9 nm for PVSK and 29.6 nm for PVSK-CF).

We performed X-ray diffraction (XRD) measurements to study the effect of $Cl_2$-CF on the crystal structure of perovskite. After the $Cl_2$-CF treatment, compared to PVSK, PVSK-$Cl_2$-CF exhibited higher XRD peak intensity, indicating improved crystallization (Supplementary Fig. 9a). Due to the ionic exchange of $I^-$ and $Cl^-$[32], XRD patterns of α-FAPbI$_3$ located at 14.00° shifted to 14.05° in PVSK-$Cl_2$-CF, indicating the formation of α-FAPbI$_{(3-x)}$Cl$_x$ (Supplementary Fig. 9b). Similarly, the XRD peak of the residual $PbI_2$ revealed a peak shift from 12.70° to 12.75°, indicating the formation of $PbI_{(2-x)}Cl_x$ (Supplementary Fig. 9c). Figure 1e illustrates the schematic diagram of the redox reaction between $Cl_2$ and perovskite. $Cl_2$ and $I^-$ ions in perovskite undergo a redox reaction to form $I_2$, and a small amount of $Cl^-$ ions enter the perovskite lattice. Therefore, the redox reaction enables Cl-doped perovskite film with larger grains.

## The distribution of Cl and the synergistic effect of HABr/$Cl_2$-CF on the perovskite film

To reveal the spatial distribution of Cl in perovskite film, XPS was first performed to detect the composition of the perovskite surface (Fig. 2a). We noticed that the Cl signals could only be detected for PVSK-$Cl_2$-CF. Subsequently, we characterized the bottom interface of

the perovskite films (Supplementary Fig. 10). Many small white particles appeared at the grain boundaries of the bottom interface of PVSK-$Cl_2$-CF, which was probably due to $Cl^-$ entering the bottom interface, forming a Pb-Cl-I mixed compound. The corresponding energy dispersive spectroscopy (EDS) results in Fig. 2b revealed that the PVSK-$Cl_2$-CF exhibited the highest Cl content of 2.64% at the bottom interface (vs. 0.28% for PVSK, 0.25% for PVSK-CF). Note that MACl was used to adjust the crystallization of perovskite, so a small amount of Cl was detected in the PVSK sample. The XPS and EDS results together with the XRD results in Supplementary Fig. 9 confirmed that $Cl_2$-CF could introduce $Cl^-$ anion into the bulk of perovskite and diffuse to the bottom side of the film.

As we know, defects are inevitable in solution-processed polycrystalline perovskite films[31,40]. To reduce those defects and improve the film quality, we passivated the perovskite films with an organic salt HABr, which can be well dissolved in CF or $Cl_2$-CF. Supplementary Fig. 11a–c and Fig. 2c show the SEM images and the statistical grain size for the control perovskite film (PVSK), HABr/CF-treated perovskite film (PVSK-HABr/CF), and HABr/$Cl_2$-CF-treated perovskite film (PVSK-HABr/$Cl_2$-CF). The PVSK exhibited small perovskite grains (black parties) with a lot of residual $PbI_2$ grains (white parties)[41] (Supplementary Fig. 11a). For the PVSK-HABr/CF in Supplementary Fig. 11b, a significant decrease in the amount of $PbI_2$ can be observed, which could be ascribed to the transformation to 2D perovskite. The similar average grain size for the PVSK and the PVSK-HABr/CF confirmed that the post-treatment didn't affect the crystal of the 3D perovskite. Supplementary Fig. 11c presents the morphology of the PVSK-HABr/$Cl_2$-CF. Along with an obvious reduction of the surface $PbI_2$, the perovskite grains significantly increased to even up to ~5 μm (Fig. 2c). The effect of $Cl_2$-CF on the morphology of perovskite film changes was further revealed by the AFM results in Supplementary Fig. 11d–f. The roughness of the PVSK-HABr/$Cl_2$-CF is largely reduced to 26.2 nm, which was comparable with that of 25.7 nm for the PVSK-HABr/CF and much less than that of 32.9 nm for the PVSK (Supplementary Fig. 11g–i). The effect of HABr/$Cl_2$-CF treatment on the $PbI_2$-free (or $PbI_2$-less) perovskite was also studied. As shown in the SEM images in Supplementary Fig. 12a, b, no distinct secondary growth of crystal grains can be observed. This result indicated that during the treatment by the HABr/$Cl_2$-CF solution, $Cl_2$ was most likely first reacted with the residual $PbI_2$ and this reaction triggered the subsequent secondary grain growth, leading to high-quality perovskite films with larger grains.

According to XRD results in Supplementary Fig. 13, all three perovskite films indicated the obvious characteristic peaks at 12.7°, 14.1°, and 28.3°, which correspond to the (100) crystal plane of $PbI_2$, the (001) and (002) crystal plane of perovskite phase, respectively. Among them, PVSK-HABr/CF and PVSK-HABr/$Cl_2$-CF display the additional characteristic peaks at ~4.0° corresponding to the Ruddlesden-Popper perovskite $HA_2FAPb_2Br_2I_5$ with a 2D structure ($n = 2$)[31]. To confirm the formed 2D phase induced by HABr, we probed the crystalline structure of perovskite films using grazing-incidence wide-angle X-ray scattering (GIWAXS). As shown in Fig. 2d–f, all perovskite films indicated the diffraction rings of $PbI_2$ and 3D perovskite phases. Meanwhile, the diffraction ring of the 2D perovskite phase ($n = 2$) can be observed in Figs. 2e, f. The integrated profiles of the GIWAXS patterns are shown in Supplementary Fig. 13. No signal of 2D perovskite could be detected for the PVSK. In contrast, there is a diffraction peak of the 2D structure at $q = 0.22$ Å$^{-1}$ for PVSK-HABr/CF and at $q = 0.25$ Å$^{-1}$ for PVSK-HABr/ $Cl_2$-CF, which is consistent with the XRD results in Supplementary Fig. 12. Especially, due to the oxidizing property of $Cl_2$-CF, a slight shift of the diffraction peak of the 2D phase to a higher angle for PVSK-HABr/$Cl_2$-CF can be ascribed to the partial replacement of $I^-$ by $Cl^-$.

To study the changes in surface elements of perovskite films, we conducted XPS measurements. Only the PVSK-HABr/$Cl_2$-CF exhibits

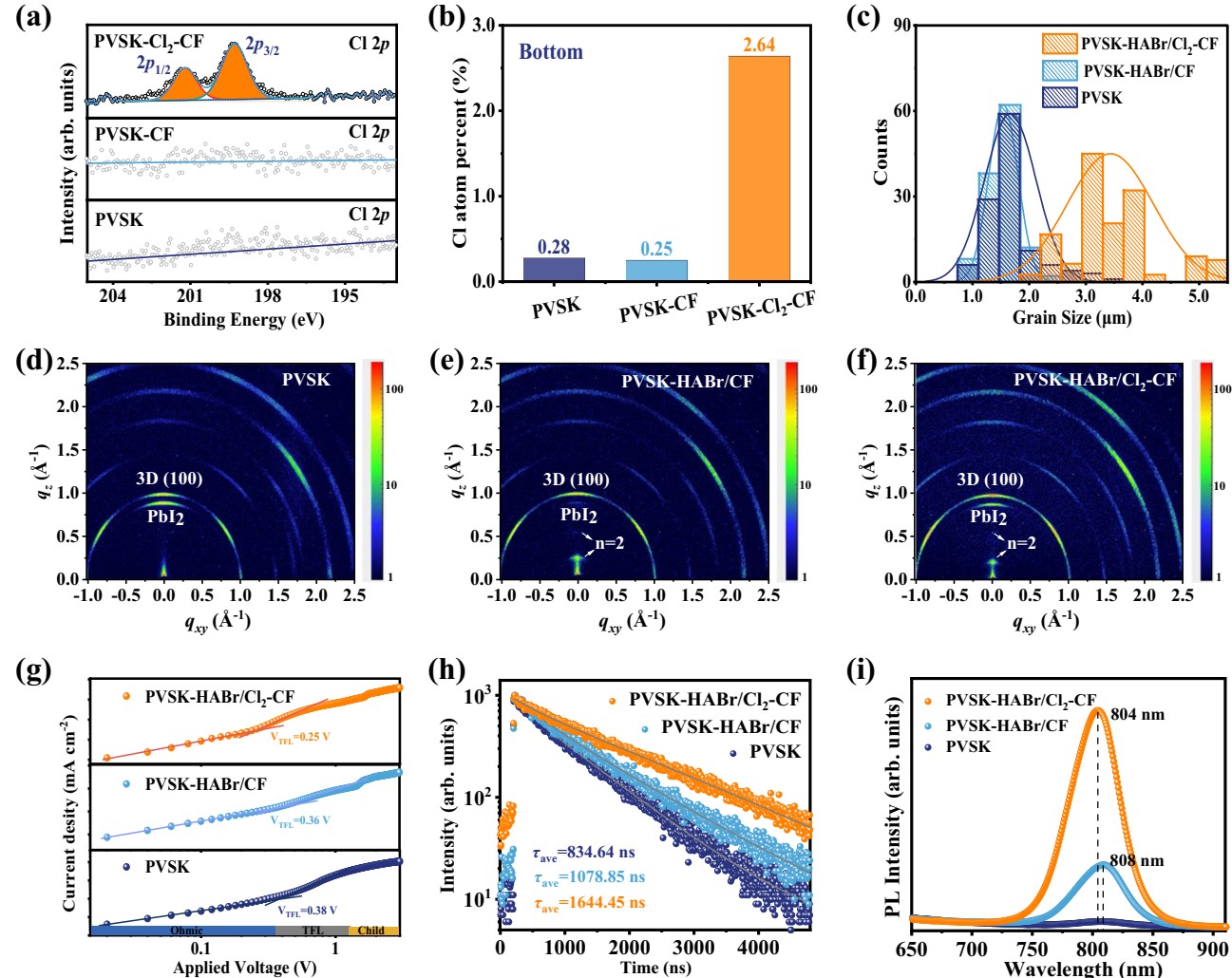

**Fig. 2 | The distribution of Cl and the synergistic effect of HABr/Cl₂-CF on the perovskite film. a** Cl 2*p* XPS spectra of PVSK, PVSK-CF, and PVSK-Cl₂-CF. **b** Cl content at the bottom surface of perovskite films. **c** Grain size distribution of PVSK, PVSK-HABr/CF, and PVSK-HABr/Cl₂-CF. **d-f** Grazing-incidence wide-angle X-ray scattering (GIWAXS) characterization with a grazing angle of 0.5°. **g** The space-charge limited-current (SCLC) measurements of the electron-only devices. **h** Time-resolved photoluminescence (TRPL) and **i** steady-state photoluminescence (PL) spectra of perovskite films.

the Cl signal with a Cl $2p_{1/2}$ peak located at 199.9 eV and a Cl $2p_{3/2}$ peak located at 198.3 eV (Supplementary Fig. 15a), which suggests the incorporation of Cl. As shown in Supplementary Fig. 15b, the PVSK and PVSK-HABr/CF present a similar Pb signal with a Pb $4f_{5/2}$ peak located at 143.2 eV and a Pb $4f_{7/2}$ peak located at 138.3 eV. However, the Pb $4f$ peaks for PVSK-HABr/Cl₂-CF exhibited a shift to the high binding energy. This shift is probably due to the formation of the Pb-Cl bond in PVSK-HABr/Cl₂-CF[42].

To investigate whether the formed Br₂ in the HABr/Cl₂-CF solution affected the perovskite film, we started by studying the state change of Cl₂ in the HABr/Cl₂-CF solution (Supplementary Fig. 16a, b). When placing the wet starch potassium iodide test paper above the HABr/Cl₂-CF solution, it turned blue immediately. This result indicated that although partial Cl₂ took part in oxidizing the Br⁻ ions of HABr to be Br₂, there was residual Cl₂ in the solution. Therefore, compared to the Cl₂-CF solvent, Cl₂ in the HABr/Cl₂-CF solution can be divided into two parts. One part took part in oxidizing Br⁻ ions to Br₂ and left Cl⁻ ions in the solution. The other part was the residual Cl₂ in Cl₂-CF solvent, which would penetrate the depth of perovskite films to trigger the redox reaction with perovskite, inducing the Cl-doping and secondary growth of perovskite crystal grains. During the post-treatment experiment using HABr/Cl₂-CF solution, we observed a

light-yellow color for the as-treated perovskite films (Supplementary Fig. 17), which can be attributed to the existence of Br₂ on the per-ovskite surface. After annealing at 100 °C for about 10 min, the light-yellow color disappeared, which could be attributed to the volatilization of Br₂ species.

To investigate the defect-related carrier transport dynamics, we quantitatively evaluated the trap density of the perovskite films. According to the space-charge limited-current (SCLC), trap density ($N_t$) is determined by the trap-filled voltage ($V_{TFL}$) following the equation:

$$N_t = 2\varepsilon_0 \varepsilon_r V_{TFL}/qL^2 \quad (1)$$

where $\varepsilon_0$, $\varepsilon_r$, $q$, and $L$ are the vacuum permittivity, relative permittivity, elementary charge, and the thickness of the perovskite film, respectively[43,44]. Based on the electron-only devices with a structure of ITO/SnO₂/perovskite/[6,6]-phenyl-C61-butyric acid methyl ester (PCBM)/Ag, the current-voltage curves were recorded in Fig. 2g. We found that the $N_t$ decreased from $2.73 \times 10^{15}$ for the PVSK to $2.59 \times 10^{15}$ cm⁻³ for the PVSK-HABr/CF, while the PVSK-HABr/Cl₂-CF exhibited the lowest $N_t$ of $1.80 \times 10^{15}$ cm⁻³. Figure 2h shows the time-resolved photoluminescence (TRPL) for perovskite films. The curves

were fitted with a biexponential function:

$$y = y_0 + A_1 e^{(-x/\tau_1)} + A_2 e^{(-x/\tau_2)} \tag{2}$$

the average carrier lifetime was determined by:

$$\tau_{avg} = (A_1 \tau_1^2 + A_2 \tau_2^2)/A_1 \tau_1 + A_2 \tau_2 \tag{3}$$

where the fast decay process ($\tau_1$) is related to the nonradiative recombination that occurred at the perovskite surface and grain boundaries, and the slow decay process ($\tau_2$) is related to the non-radiative recombination of photo-generated free carriers in bulk perovskite[45]. The fitted results (Supplementary Table 1) showed that, compared with an average lifetime of 834.6 ns for the PVSK, the PVSK-HABr/CF exhibited a longer average lifetime of 1078.9 ns, while the PVSK-HABr/Cl$_2$-CF displayed the longest average lifetime of 1644.5 ns. Figure 2i shows the steady-state photoluminescence (PL) results of perovskite films. Among the three perovskite films, the peak intensity of PVSK-HABr/Cl$_2$-CF is the strongest, indicating the suppressed nonradiative recombination. In addition, the blue shift of the emission peak suggests a widening bandgap of the PVSK-HABr/Cl$_2$-CF due to the introduction of Cl$^-$ into the perovskite lattice, which can be confirmed by the Tauc results in Supplementary Fig. 18. The bandgap of the PVSK and PVSK-HABr/CF is about 1.54 eV, while that of the PVSK-HABr/Cl$_2$-CF increased to 1.55 eV. The 2D PL mapping images shown in Supplementary Fig. 19a–c demonstrated that PVSK-HABr/Cl$_2$-CF showed overall higher PL intensity as well as larger perovskite grains, indicating better passivation effect and second grain growth. Moreover, there were a lot of dark grain boundaries for the PVSK and PVSK-HABr/CF samples while most of them disappeared in the PVSK-HABr/Cl$_2$-CF sample. These results indicated that the HABr/Cl$_2$-CF treatment led to a superior passivation effect at grain boundaries.

Based on the abovementioned changes in morphology and crystal structure, the role of HABr/CF and HABr/Cl$_2$-CF post-treatment is illustrated in Supplementary Fig. 20. The PVSK contains many small perovskite grains that could generate many grain boundaries and numerous defects, trapping the photo-generated carriers (Supplementary Fig. 20a). The introduction of HABr dissolved in CF can form a 2D/3D structure to passivate the defects in perovskite films (Supplementary Fig. 20b). However, this enhancement is insufficient to substantially improve the quality of perovskite films. Encouragingly, post-treatment with HABr dissolved in Cl$_2$-CF can realize multiple functions (Supplementary Fig. 20c). Except for the formed 2D/3D structure, the secondary growth of perovskite grains induced by Cl$_2$-CF can largely reduce the number of grain boundaries and defects. Therefore, we believe that using HABr/Cl$_2$-CF to treat perovskite films can more effectively improve the perovskite film quality by a synergistic effect that HABr induces the 2D/3D structure, and Cl$_2$-CF results in the secondary growth and Cl-doping of the 3D perovskite film.

## Defect physics and carrier transport of PSCs based on PVSK-HABr/Cl$_2$-CF

We employed the trap density of states (tDOS) measurement to evaluate defect states in perovskite films. Previous studies claimed that the deep trap states are mainly related to the surface defects of perovskite films and the shallow trap states are likely resulted from the bulk perovskite films[46,47]. Figure 3a shows that the PVSK-HABr/Cl$_2$-CF device displayed the lowest tDOS in the deep-trap region (0.40-0.55 eV), suggesting the high-quality PVSK-HABr/Cl$_2$-CF with much-suppressed defects.

Electrochemical impedance spectroscopy (EIS) was measured at a bias of 0.9 V under dark conditions, and the results were fitted with an equivalent circuit shown in the inset of Fig. 3b. For the double arc shape in the Nyquist plot, the low-frequency arc is associated with the perovskite dielectric relaxation resistance ($R_{dr}$), whereas the high-frequency one is associated with the drift-diffusion and recombination processes[48]. The PVSK-HABr/CF device displayed a larger perovskite recombination resistance ($R_{rec}$) of 1800 Ω than that of PVSK device (1500 Ω), while PVSK-HABr/Cl$_2$-CF device shows the largest $R_{rec}$ of over 2000 Ω, which can effectively hinder the recombination between electrons and holes. Transient photovoltage measurement (TPV) shows that the decay lifetime of PVSK-HABr/CF device (0.88 ms) was

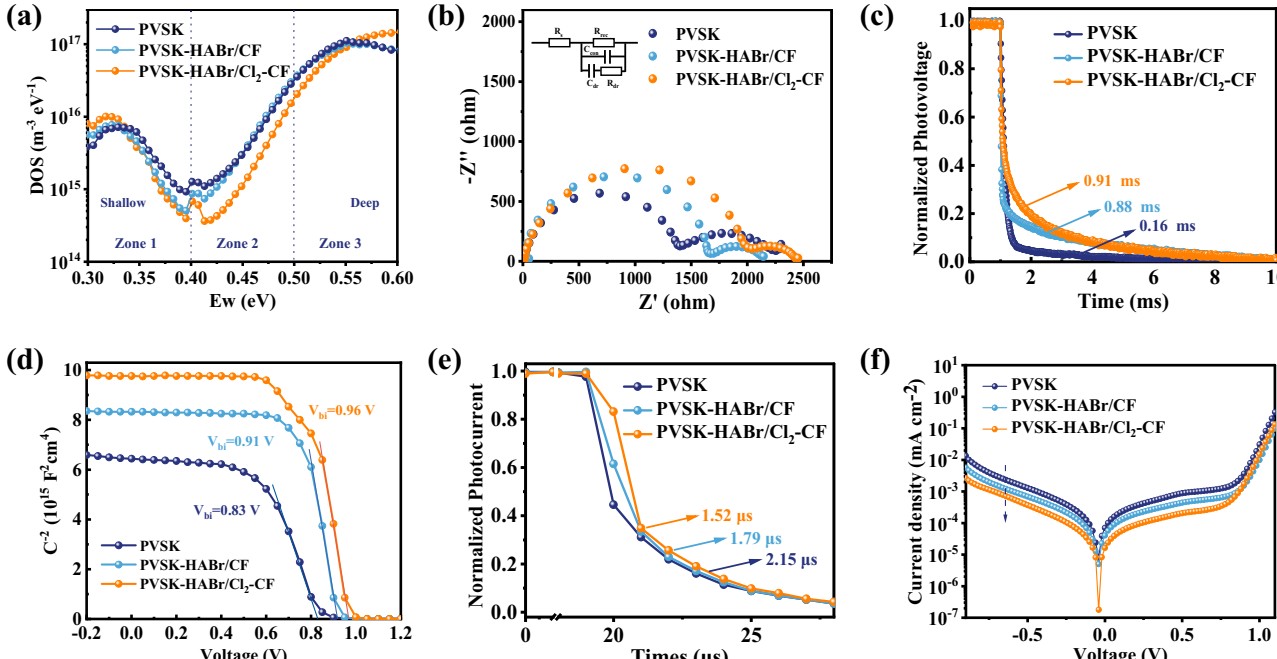

**Fig. 3 | Defect physics and carrier transport in PSCs based on the PVSK, PVSK-HABr/CF, and PVSK-HABr/Cl$_2$-CF. a** Trap density of states (tDOS). **b** Electrochemical impedance spectroscopy (EIS) at a bias of 0.9 V under dark conditions. **c** Transient photovoltage measurements (TPV) at the open circuit. **d** Mott-Schottky analysis. **e** Transient photocurrent (TPC) at the short circuit. **f** $J$-$V$ curves measured under dark conditions.

higher than that of PVSK device (0.16 ms), while PVSK-HABr/Cl₂-CF device showed the highest decay lifetime (0.91 ms), further indicating that the recombination in PVSK-HABr/Cl₂-CF device was greatly suppressed (Fig. 3c).

From the Mott-Schottky analysis[49], the obtained built-in potential ($V_{bi}$) of the PVSK, PVSK-HABr/CF, and PVSK-HABr/Cl₂-CF devices were 0.83, 0.91, and 0.96 V, respectively (Fig. 3d). It indicates that the PVSK-HABr/Cl₂-CF device has a stronger internal driving force for the separation and transport of charge carriers. Transient photocurrent (TPC) measurements were employed to study carrier transport. As shown in Fig. 3e, the charge transfer time ($\tau$) was decreased from 2.15 µs to 1.79 and 1.52 at the short circuit, indicating that PVSK-HABr/Cl₂-CF device has the fastest carrier extraction capability, which is beneficial to reduce the hysteresis. Figure 3f shows the dark current of the three devices. The dark current of PVSK-HABr/Cl₂-CF device is much lower than PVSK and PVSK-HABr/CF devices, indicating suppressed leakage pathways and improved ideality.

## Enhancing the photovoltaic performance by HABr/Cl₂-CF post-treatment

The HABr/Cl₂-CF post-treatment has a significant effect on the photovoltaic performance of PSCs. We fabricated photovoltaic devices with the structure of ITO/SnO₂/perovskite/2,2′,7,7′-tetrakis(N,N-di(4-methoxyphenyl)amino)-9,9-spirobifluorene (Spiro-OMeTAD)/Ag (Fig. 4a). The discontinuous white particles between the perovskite layer and hole transfer layer in the PVSK-based PSC are deemed as PbI₂ (Supplementary Fig. 21). Under long-term operating conditions, the unreacted PbI₂ at the carrier transport interface is essentially the catalytic site that triggers perovskite decomposition[50]. In the PVSK-HABr/CF and PVSK-HABr/Cl₂-CF device, the amount of PbI₂ was decreased due to the transformation to 2D perovskite. Moreover, the PVSK-HABr/Cl₂-CF device displayed vertically arranged perovskite grains, which was beneficial to charge transport.

Figure 4b showed the $J$-$V$ curves under the reverse scan direction (from the open circuit to the short circuit). It demonstrated that the open-circuit voltage ($V_{OC}$) of PVSK-HABr/Cl₂-CF was increased from

1.116 of PVSK and 1.147 of PVSK-HABr/CF to 1.168 V, while the short-circuit current density ($J_{SC}$) didn't distinctly change. The statistical photovoltaic performances are presented in Fig. 4c, d, and Supplementary Fig. 22. Compared to the PVSK devices, PVSK-HABr/CF devices indicate a minor increase in PCE, while the PVSK-HABr/Cl₂-CF devices indicated a large increment in PCE based on the improved $V_{OC}$. After optimization, we obtained a champion PCE of 24.21% ($V_{OC}$ of 1.168 V, $J_{SC}$ of 25.40 mA cm⁻², and FF of 81.58%) for the PVSK-HABr/Cl₂-CF device. Supplementary Fig. 23 shows the incident photon-to-current conversion efficiency (IPCE) in 300-900 nm. The integrated $J_{SC}$ for the PVSK, PVSK-HABr/CF and PVSK-HABr/Cl₂-CF was 24.78, 24.77, and 24.62 (mA cm⁻²), respectively. It is well matched with the value obtained from the $J$-$V$ curves (<3% discrepancy), proving the reliability of the $J_{SC}$ results.

To further investigate the effect of Cl₂, we compared the performance of the devices based on perovskite post-treatment by n-HACl dissolved in pure chloroform without Cl₂, n-HACl dissolved in Cl₂-dissolved chloroform, and n-HABr dissolved in Cl₂-dissolved chloroform. As shown in Supplementary Fig. 24, n-HACl and n-HABr dissolved in Cl₂-dissolved chloroform exhibited better average $V_{OC}$ and PCE than n-HACl dissolved in fresh chloroform, indicating that the dissolved Cl₂ played an important role. This was most likely due to Cl₂-induced secondary crystal growth that further reduced the defects in the bulk of the 3D perovskite.

To investigate whether the formed Br₂ in the HABr/Cl₂-CF solution affected the passivation effect, we added different mass ratios of Br₂ into the solution of HABr dissolved in CF. Note that no Br₂ can be produced in the PVSK and HABr/CF due to there being no Cl₂ in the CF. So the effect of the added Br₂ can be studied. The statistics of the devices were shown in Supplementary Fig. 25. It demonstrated that Br₂ did not improve the $V_{OC}$ and FF of the device. Although Br₂ can also oxidize I⁻, the effect of Br₂ on the device performance was very different from that of Cl₂. This can be attributed to the weaker oxidability and lower penetration ability due to the higher boiling point of Br₂. As indicated by the color change after spin-coating, most Br₂ may remain on the surface of perovskite film and give litter effects. As expected,

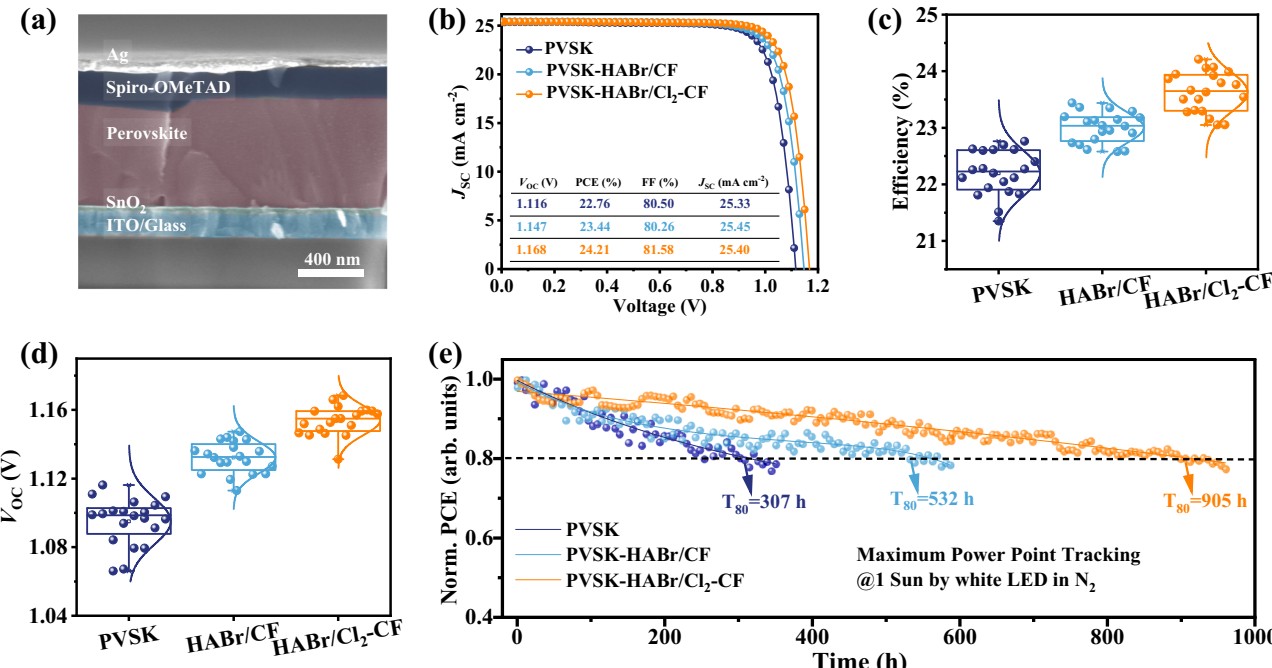

**Fig. 4 | Photovoltaic performance of the PSCs based on PVSK, PVSK-HABr/CF, and PVSK-HABr/Cl₂-CF. a** Cross-sectional SEM image of the n-i-p perovskite solar cells. **b** $J$-$V$ curves measured in the reverse scan direction (1.2 to 0 V, 250 mV s⁻¹). **c** Statistical PCE. **d** Statistical $V_{OC}$. **e** Maximum power point tracking of PSCs under a LED lamp with a light intensity of 100 mW cm⁻². 'Normalized' is denoted as 'Norm'.

along with the introduction of HABr/Cl$_2$-CF, the hysteresis became negligible, which suggested that the defects were greatly passivated by HABr/Cl$_2$-CF treatment (Supplementary Fig. 26). Furthermore, the steady-state PCE measured at the maximum power point ($V_{max}$) is shown in Supplementary Fig. 27. Compared with the other two devices, the PVSK-HABr/Cl$_2$-CF device exhibited the most stable PCE of 23.30% at 0.99 V after measuring for 600 s.

Stability is critical to PSCs technology[51–54]. We investigated the long-term operational stability of the PSCs. The unencapsulated PVSK-HABr/Cl$_2$-CF device presented excellent operational stability under the maximum power point (MPP) tracking at 1-sun illumination with a LED[43]. As shown in Fig. 4e, the PVSK-HABr/Cl$_2$-CF device can obtain a $T_{80}$ lifetime (the time when the efficiency decreased to 80% of the initial PCE) of 905 h, which is much longer than the PVSK device (307 h) and PVSK-HABr/CF device (532 h), suggesting the dramatical enhancement in the operational stability. We further tracked the ambient stability of the devices (Supplementary Fig. 28). After 1920 hours of aging in the air with a relative humidity of 10% and a temperature of ~25 °C, the unencapsulated device based on PVSK-HABr/Cl$_2$-CF maintained 91% of its original PCE while the device based on PVSK and PVSK-HABr/CF maintained 78% and 82% respectively. Supplementary Fig. 29 and Supplementary Fig. 30 showed that HABr/Cl$_2$-CF treatment improved the thermal stability. The devices based on PVSK-HABr/Cl$_2$-CF remained over 92% of the initial PCE after heating for 400 hours at 55 °C in a nitrogen environment and remained over 81% of the initial PCE after heating for 106 hours at 85 °C in a nitrogen environment.

## Discussion

In conclusion, we report in-situ formed oxidative Cl$_2$-dissolved chloroform as the post-processing solvent for bulky cations to construct 2D/3D perovskite heterojunctions, which allows us to obtain high-quality perovskite films with passivated surface and enlarged grains in the bulk. The introduced Cl$^-$ further diffuses to passivate the buried interface of perovskite solar cells. These effects are enabled by the redox reaction between Cl$_2$ and I$^-$. As a result, the defect density of the perovskite films can be reduced and the nonradiative recombination can be largely suppressed. Finally, we achieve a high PCE of 24.21% for the optimized HABr/Cl$_2$-CF device with negligible hysteresis effect. The optimized device also showed a substantially enhanced long-term operational stability with a $T_{80}$ lifetime of 905 h.

## Methods

### Materials

Unless otherwise stated, all chemicals were purchased from Sigma-Aldrich and used as received.

### Synthesis of chlorine-dissolved chloroform (Cl$_2$-CF)

A newly purchased chloroform (HPLC-grade) is stored in a nitrogen environment for isolating from H$_2$O and O$_2$. For obtaining the Cl$_2$-CF, fresh CF was transferred to a transparent bottle and illuminated with a xenon lamp (Abet Technologies' model 11002 SunLite$^{TM}$ Solar Simulator, 100 mW cm$^{-2}$) for 24 h in the ambient air with ~35% relative humidity. During the illumination process, we periodically detected the state of the CF with wet starch potassium iodide test paper. Once the wet starch potassium iodide test paper turns blue, it suggests that the fresh CF has been translated into Cl$_2$-CF.

### Film deposition and device fabrication

The ITO-coated glasses were washed in deionized water, acetone, IPA, and ethanol for 20 minutes respectively by ultrasonic treatment. After drying, the surface was treated with plasma (Harrick, PDC-002-HP) for 5 min. Then, the SnO$_2$ precursor (Alfa Aesar, 15% in H$_2$O colloidal dispersion, diluted to 5%) was deposited onto the ITO substrate by spin-coating (4000 rpm for 20 s). Later, these samples were annealed at

150 °C for 15 min. After the ITO/SnO$_2$ substrates were cooled to room temperature, another plasma treatment process was performed for 5 min to clean the SnO$_2$ film surface. Subsequently, the SnO$_2$-coated substrates were transferred to a glovebox with an N$_2$ atmosphere, and perovskite films were prepared via a two-step sequential deposition method. Firstly, PbI$_2$ solution was prepared by dissolving PbI$_2$ (691.5 mg, TCI) and 5% mole ratio of CsI (relative to PbI$_2$, 19.5 mg) in 1 mL mixed solvent of DMF (N, N-dimethylformamide) and DMSO (dimethylsulfoxide) with a v/v ratio of 9:1. PbI$_2$ film was obtained by spin-coating (2000 rpm for 30 s) the PbI$_2$ solution onto the substrates and annealing at 70 °C for 60 s. Secondly, the salts solution was mixed with FAI (118.6 mg, Dyesol), MACl (18 mg, Dyesol), MABr (5.6 mg, Dyesol), and MAI (10 mg, Dyesol) in 2 mL of IPA. It was spin-coated onto the PbI$_2$ film (1,800 rpm for 30 s) and annealed at 150 °C for 15 min in the air (relative humidity is about 35%). Based on the above steps, control perovskite films (PVSK) were obtained. After the preparation of the perovskite layer, HABr/CF (or HABr/Cl$_2$-CF) (10 mmol mL$^{-1}$) was spin-casted at 5500 rpm for 30 s on top of the perovskite films and annealed at 100 °C for 10 min in N$_2$. The hole transporting layer (HTL) solution was prepared by dissolving 90 mg of Spiro-OMeTAD into 1 mL of chlorobenzene (CB), followed by the addition of 4-tert-butylpyridine (28.8 µL), and bis(trifluoromethane) sulfonamide lithium salt (17.5 µL, 520 mg mL$^{-1}$ in acetonitrile). It was spin-coated onto the perovskite films (3000 rpm for 30 s). Finally, about 60 nm of Ag was thermally evaporated on top of the Spiro-OMeTAD layer. For the thermal stability test, a polymer (PBDB-T)-doped Spiro-OMeTAD hole-transporting layer was employed to improve the thermal stability of the devices.

### The method to get the buried interface of perovskite film without damage

To expose and characterize the buried interface without damaging the perovskite film, the previously reported technique was used[55]. Briefly, perovskite samples with the structure of ITO/PTAA/perovskite/Ag were first fabricated. And then the samples were immersed in CB for 20 min in the N$_2$-filled glovebox. After the PTAA layer was dissolved by CB, the perovskite/Ag film detached from the substrate and floated on the surface of CB. Finally, the lift-off perovskite film was fixed to a base with its bottom side.

### Calculation of $N_t$ ($E_\omega$)

The tDOS ($N_t$ ($E_\omega$)) of the device was determined by measuring the impedance spectroscopy (EIS) and the Mott-Schottky curves in the dark using the previously reported method[56].

The $N_t$ can be estimated by Eq. (4):

$$N_t(E_\omega) = -\frac{\omega}{K_B T} \times \frac{V_{bi}}{eW} \times \frac{dC}{d\omega} \tag{4}$$

where $\omega$ was the angular frequency, $K_B$ was the Boltzmann constant, $T$ was the temperature, $V_{bi}$ was the built-in electric field, $e$ was the electron charge, $W$ was the depletion width and $C$ was the capacitance. The independent variable of energy $E_\omega$ can be determined by Eq. (5):

$$E_\omega = K_B T \times \ln \frac{2\beta_\rho N_v}{\omega} \tag{5}$$

where $\beta_\rho$ was the capture coefficient of hole, $N_v$ was the effective density of states in the valence band. The depletion layer width $W$ can be calculated by Eq. (6):

$$W = \sqrt{\frac{2\varepsilon_s}{q} \times \frac{N_A + N_D}{N_A \times N_D} \times V_{bi}} \tag{6}$$

where $\varepsilon_s$ was the dielectric constant of perovskite active layer, $N_A$ and $N_D$ were the doping concentrations of the hole-transporting layer and the electron-transporting layer respectively. $V_{bi}$ can be determined by measuring the Mott-Schottky curve and calculated by Eq. (7):

$$\frac{1}{C^2} = \frac{2}{n^2} \times \frac{1}{q\varepsilon_s} \times \frac{N_A + N_D}{N_A \times N_D}(V - V_{bi}) \quad (7)$$

where $n$ was a constant of proportionality. Then the formula for $k$ was expressed as:

$$k = \frac{2}{n^2} \times \frac{1}{q\varepsilon_s} \times \frac{N_A + N_D}{N_A \times N_D} \quad (8)$$

Therefore, Eq. (6) was simplified to Eq. (9).

$$W = \varepsilon_s \times \sqrt{-kn^2 \times V_{bi}} \quad (9)$$

In addition, the geometric capacitance $C_g$ can be obtained from the high-frequency region of the EIS data by Eq. (10):

$$C_g = \frac{n\varepsilon_s}{d} \quad (10)$$

where $d$ was the thickness of the perovskite layer, then Eq. (9) can again be expressed as:

$$W = C_g \times d \times \sqrt{-kV_{bi}} \quad (11)$$

Therefore, $\beta_p$, $N_v$, $\varepsilon_s$, $C_g$, d, k, $V_{bi}$, and $\frac{dC}{d\omega}$ of related materials should be known to calculate $N_t$. The value of $10^{-8}$ cm$^3$/s for $\beta_p$ and the value of $2.524 \times 10^{-19}$ /cm$^3$ for $N_v$ was used in the calculation[56].

### Characterization

UV-vis spectrophotometer (Techcomp, UV2600) was used to study the chemical properties of the involved solution. Grazing incidence wide-angle X-ray scattering (GIWAXS) measurements were performed on the Xeuss Sax/WAXS system (Xenocs, France) with a Pilatus3R 300 K detector (grazing angle of 0.5°). Steady-state photoluminescence (PL) spectra of perovskite films were obtained on an instrument provided by Xipu Electronics with integrated spheres installed in a glovebox. The scanning electron microscope (SEM) images of all samples were observed by JEOL, JSM-7610F. XRD was recorded by SmartLab X-ray diffractometer (Rigaku Corporation) with the Cu $K_\alpha$ radiation source. The surface morphology of perovskite films was studied using AFM on a multi-mode 8 SPM system (Bruker). Time-resolved photoluminescence (TRPL) was measured by FLS920 (Edinburgh Instruments Limited) with a 375 nm pulse excitation. The CHI660E electrochemical workstation was used to collect the dark J-V curves, electrochemical impedance spectroscopy (EIS), and space-charge-limited current (SCLC). Current density-voltage characteristics were measured using a source meter (Keithley 2400) under AM 1.5 G conditions (EnliTech, AAA solar simulator). Light intensity was calibrated using an NREL-calibrated silicon solar cell equipped with an infrared cut-off filter (KG-5). A scan rate of 250 mV s$^{-1}$ (voltage step of 10 mV, delay time of 40 ms) was used. The active area of PSCs was 0.2 cm$^2$. A black shadow mask was used to define an effective area of 0.12 cm$^2$ for the measured PSCs. IPCE spectra were measured in DC mode using the QE-R666 system (Enli tech). The mott-Schottky analysis, trap density of states (tDOS), transient photocurrent (TPC), and transient photovoltage (TPV) were performed on Zahner's electrochemical workstation equipped with a transient electrochemical measurement unit (Fast CIMPS). The operational stability of the best-performing PSCs was tested on a solar cell stability test system (Suzhou D&R

Instruments Co, Ltd.) under 100 mW cm$^{-2}$ illumination using a white LED light source. The temperature of the sample was about 60 °C. PL mapping images were acquired using a laser confocal microscope (Leica TCS SP8) excited at a 488 nm pulse.

### Reporting summary

Further information on research design is available in the Nature Portfolio Reporting Summary linked to this article.

## Data availability

The data that support the findings of this study are available in the following repository: https://doi.org/10.6084/m9.figshare.22698475. The source data is provided with this work. Source data are provided with this paper.

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

## Acknowledgements

Y.L. and K.L. contributed equally to this work. This work was financially supported by the National Natural Science Foundation of China (22179042, L.X.; U21A2078, Z.W.; and 51902110, C.T.), and the Natural Science Foundation of Fujian Province (2020J06021, Z.W.; and 2020J01064, C.T.). We thank the comprehensive experiment center at Huaqiao University for providing various tests.

## Author contributions

Z.W. and L.X. supervised the work. Y.Lu. and K.L. fabricated devices and analyzed the data. Y.Lu., L.Y., and Z.F. performed EIS measurements. W.F., L.Z., and Y.J. performed UV-vis absorption spectroscopy measurements. Y.Lu., K.L., and L.S. performed TPV and TPC measurements. Y.Lu., P.S., and K.L. performed MPP measurements. C.T., P.X., W.T., and Y.Li. contributed to the data analysis. Y.Lu., K.L., and L.X. co-wrote the paper. Z.W., L.X., K.L., and Y.Lu. revised the paper. All authors read and commented on the paper.

## Competing interests

The authors declare no competing interests.
