## [Peer Review File · Nature Communications]

Dissolved-Cl₂ Triggered Redox Reaction Enables
High-performance Perovskite Solar CellsREVIEWER COMMENTS

Reviewer #1 (Remarks to the Author):

In the manuscript entitled “Dissolved-Cl₂ Triggered Redox Reaction Enables High-performance Perovskite Solar Cells”, Luo et al. show an effective method to improve perovskite cell efficiency and stability by combining CF-Cl₂ and n-HABr to treat the perovskite surface. The paper presented some insights into the mechanism however I am not quite convinced that the novelty is sufficient for publication in Nature Communications. In addition, the efficiency is 24.06% (not a steady state efficiency), which substantially lags behind state-of-the-art PSCs (25.6%). Therefore, I think it can be more suitable for more PV-specialized journal. Some other comments are below.

In the abstract, the authors claim that “previous reports only physically deposit a 2D perovskite passivation layer on the 3D perovskite layer. These methods are limited to surface passivation only, and the bulk 3D perovskite remains defective”. I think this claim is not correct since several reports also experimented with adding 2D into the perovskite bulk and achieved some promising results.

The XRD peak shift from 14.07 to 14.09 is very small? Is it within the resolution of the equipment?

It seems like the device performance is enhanced due to the synergetic effect of Cl⁻ diffusion and n-HABr passivation. What’s about using n-HACl passivation? Would it have similar effect?

There is no Cl in the control perovskite composition, why did the authors observe up to 0.25% Cl content using XPS? How did the author characterize the buried interface without damaging the perovskite film? This point needs to be clearly mentioned in the manuscript.

How did the author measure the tDOS in the device? This need to be explained in the manuscript.

In terms of stability, the authors only demonstrate enhanced operation stability. The authors need to demonstrate ambient stability and thermal stability as well.

Reviewer #2 (Remarks to the Author):

This paper reports that chloroform is oxidized to form Cl₂, and the Cl₂ formed in this way is treated on the surface of perovskite to penetrate not only the surface but also the bulk, effectively passivating defects, and growing crystal grains. The approach is interesting and the influence of Cl₂ is clear from the experimental results, but there are critical concerns about reaching conclusions. I believe that only when these concerns are addressed can we decide whether to publish the paper.

1. [p6. Line 127-135] In the manuscript, the authors described that the Cl incorporation shifts the peak of FAPbI₃ from 14.07 to 14.09 but Supplementary Fig.9a shows the opposite shift from 14.07 to 14.04.

The peak of PbI₂ is too. Data in Supplementary Fig 9a reveals that Cl₂-CF treatment increases the lattice parameter of FAPbI₃ which is in contrast to the substitution of smaller Cl ions into I sites in FAPbI₃. This is critical because it is the opposite of the main insist of this paper.

2. The authors insist that Cl₂ plays an important role in the passivation and grain growth of FAPbI₃ film which is confirmed by analysis in Figure 1. In addition, the authors insist that in the HABr/Cl₂-CF treatment, Cl₂ plays the same role as the Cl₂-CF treatment. But as shown in Supplementary Fig6b, when HABr is dissolved in Cl₂-CF, Cl₂ produces Br₂. If the state of Cl₂ in the HABr/Cl₂-CF solution is different from the Cl₂-CF solution, the effect of Cl₂ could not be interpreted by only diffusion into the bulk. There could be an effect of Br₂.

3. Since this work fabricates FAPbI₃ by sequential method, the control film includes a lot of PbI₂. This is unusual. In most recent papers related to FAPbI₃, the thin film surface in SEM images show clear without PbI₂ which is in contrast to this work in Supplementary Fig. 11. In this paper, it is difficult to exclude the effect of the removal of PbI₂ during post-treatment, making the argument of the paper unclear. Therefore, experiments on complete FAPbI₃ thin films without PbI₂ are essential to clarify the conclusion of this paper. Complete FAPbI₃ thin films can be produced by either the two-step method [DOI: 10.1126/science.aaa9272, DOI: 10.1126/sciadv.abe3326] or by one-step method[<https://doi.org/10.1038/nature14133>].

Reviewer #3 (Remarks to the Author):

Comments to the Author

In this manuscript, authors report a new interface passivation strategy using solvent engineering method. Through Cl₂-dissolved CF, authors provide the effective construction of the 2D/3D perovskite heterojunction. Also, this method enables high efficiency of over 24% with enhanced operational stability. Additionally, authors have performed appropriate analyzes to support their claims for reduced defect with this new passivation strategy. This new passivation strategy is obviously novel and different from the papers published so far. Thus, I recommend that this manuscript is published in Nature communications. Some aspects of the work may deserve the authors' further attention and then lead to revision:

1) Clearly, this strategy will give different passivation effects at grain and grain boundary. Therefore, 2D PL mapping analysis could further enhance the quality of this manuscript.

2) The grain size through SEM appears larger, but the FWHM of XRD does not seem to change significantly. Please compare the FWHM of the XRD to make sure the grain size of bulk is increased, not just the surface, and calculate the grain size from FWHM.

3) This manuscript employs HABr to form 2D/3D structures. Therefore, the papers introducing HABr for the first time should be cited (DOI: 10.1039/C9EE00751B) and explained with its advantages.

Reviewer #1

In the manuscript entitled “Dissolved-Cl₂ Triggered Redox Reaction Enables High-performance Perovskite Solar Cells”, Luo et al. show an effective method to improve perovskite cell efficiency and stability by combining CF-Cl₂ and n-HABr to treat the perovskite surface. The paper presented some insights into the mechanism however I am not quite convinced that the novelty is sufficient for publication in *Nature Communications*. In addition, the efficiency is 24.06% (not a steady state efficiency), which substantially lags behind state-of-the-art PSCs (25.6%). Therefore, I think it can be more suitable for more PV-specialized journal. Some other comments are below.

Response: Thanks for the reviewer’s critical comments. We are sorry that the novelty of the work was probably not clearly expressed in the previous manuscript. We would like to highlight the findings of this work as follows. Firstly, this work reported a novel one-step post-treatment strategy that can passivate the perovskite surface and enlarge the grains in the bulk of the film at the same time, which we think represents an important advance in the defect management of PSCs. Secondly, the introduction of in-situ formed oxidative Cl₂-dissolved chloroform as the post-processing solvent was first proposed in perovskite solar cells, and the mechanism of how Cl₂ affects the performance of PSCs was clarified. Thirdly, Cl₂-dissolved chloroform was used to dissolve bulky cations to construct 2D/3D perovskite heterojunctions, the redox reaction between Cl₂ and I⁻ triggered the secondary grain growth, and the introduced Cl⁻ diffuses to passivate the buried interface of perovskite solar cells. These effects led to an increase in V_{OC} , efficiency, and stability of PSCs. Based on these advances and findings, we believe this work will be of broad interest to the readership of *Nature Communications*.

The Abstract has been revised accordingly in the revised manuscript to emphasize the novelty of the work.

Abstract

Constructing 2D/3D perovskite heterojunctions is effective for the surface passivation

of perovskite solar cells (PSCs). However, previous reports that studying perovskite post-treatment only physically deposits 2D perovskite on the 3D perovskite, and the bulk 3D perovskite remains defective. Herein, we propose Cl_2 -dissolved chloroform as a multifunctional solvent for concurrently constructing 2D/3D perovskite heterojunction and inducing the secondary growth of the bulk grains. The mechanism of how Cl_2 affects the performance of PSCs is clarified. Specifically, the dissolved Cl_2 reacts with the 3D perovskite, leading to Cl/I ionic exchange and Ostwald ripening of the bulk grains. The generated Cl^- further diffuses to passivate the bulk crystal and buried interface of PSCs. Hexylammonium bromide dissolved in the novel solvent reacts with the residual PbI_2 to form 2D/3D heterojunctions on the surface. As a result, we achieved high-performance PSCs with a champion efficiency of 24.21% and substantially improved thermal, ambient, and operational stability.

Regarding the efficiencies of the devices reported in this work, we are sure that the performance has a distinct gap compared with the world record (the record is 25.8% now). However, we think the devices still belong to the mainstream high-performance PSCs, and the reported device performance would not weaken the argument of the work.

Moreover, during the revision of the manuscript, the champion PCE of this work was updated to 24.21%. In the revised manuscript, the hysteresis of the device, the IPCE spectra, the steady-state power output, and the statistical PCE have been updated accordingly.

We sincerely invite the reviewer to reconsider our revised manuscript for publication in *Nature Communications*.

Updated photovoltaic performances

Fig. 4. Photovoltaic performance of the PSCs based on PVSK, PVSK-HABr/CF, and PVSK-HABr/Cl₂-CF. (b) $J-V$ curves measured in the reverse scan direction (1.2 to 0 V, 250 mV s⁻¹). (c) Statistical PCE. (d) Statistical V_{oc} .

Supplementary Fig. 22 | Statistics of (a) FF, and (b) J_{sc} of the devices based on PVSK, PVSK-HABr/CF, and PVSK-HABr/Cl₂-CF.

Supplementary Fig. 23 | IPCE spectra and the integrated J_{sc} of the PVSK, PVSK-HABr/CF, and PVSK-HABr/Cl₂-CF based devices. (a) 300 to 900 nm, (b) 770 to 830 nm.

Supplementary Fig. 26 | J - V curves and hysteresis of the (a) PVSF, (b) PVSF-HABr/CF, and (c) PVSF-HABr/Cl₂-CF based devices.

Supplementary Fig. 27 | Steady-state power output at the maximum power point of PVSF, PVSF-HABr/CF, and PVSF-HABr/Cl₂-CF based devices.

Changes in the manuscript:

Change 1: As a result, the champion PSC delivered a high efficiency of 24.21% with negligible hysteresis.

Change 2: Fig. 4b showed the J - V curves under the reverse scan direction (from the open circuit to the short circuit). It demonstrated that the open-circuit voltage (V_{oc}) of PVSF-HABr/Cl₂-CF was increased from 1.116 of PVSF and 1.147 of PVSF-HABr/CF to 1.168 V, while the short-circuit current density (J_{sc}) didn't distinctly change. The statistical photovoltaic performances are presented in Fig. 4c, d, and Supplementary Fig. 22. Compared to the PVSF devices, PVSF-HABr/CF devices indicate a minor increase in PCE, while the PVSF-HABr/Cl₂-CF devices indicated a large increment in PCE based on the improved V_{oc} . After optimization, we obtained a champion PCE of 24.21% (V_{oc} of 1.168 V, J_{sc} of 25.40 mA cm⁻², and FF of 81.58%) for the PVSF-HABr/Cl₂-CF device. Supplementary Fig. 23 shows the incident

photon-to-current conversion efficiency (IPCE) in 300-900 nm. The integrated J_{SC} for the PVS-K, PVS-K-HABr/CF, and PVS-K-HABr/Cl₂-CF was 24.78, 24.77, and 24.62 (mA cm⁻²), respectively. It is well matched with the value obtained from the J - V curves (<3% discrepancy), proving the reliability of the J_{SC} results. As expected, along with the introduction of HABr/Cl₂-CF, the hysteresis became negligible, which suggested that the defects were greatly passivated by HABr/Cl₂-CF treatment (Supplementary Fig. 26). Furthermore, the steady-state PCE measured at the maximum power point (V_{max}) is shown in Supplementary Fig. 27. Compared with the other two devices, the PVS-K-HABr/Cl₂-CF device exhibited the most stable PCE of 23.30% at 0.99 V after measuring for 600 s.

Change 3: Finally, we achieve a high PCE of 24.21% for the optimized HABr/Cl₂-CF device with negligible hysteresis effect.

Change 4:

Entry for the Table of Contents

We adopt Cl₂-dissolved chloroform as a multifunctional and reactive solvent to construct 2D/3D perovskite heterojunctions. The redox reaction between Cl₂ and I⁻ leads to chloride doping, secondary growth of perovskite grains, and the formation of 2D/3D perovskite heterojunction. As a result, the device based on high-quality perovskite shows a champion efficiency of 24.21%.

Comment 1. In the abstract, the authors claim that “previous reports only physically deposit a 2D perovskite passivation layer on the 3D perovskite layer. These methods are limited to surface passivation only, and the bulk 3D perovskite remains defective”.

I think this claim is not correct since several reports also experimented with adding 2D into the perovskite bulk and achieved some promising results.

Response: Thanks for this important comment. The previous claim was indeed not accurate and led to misunderstanding. We intended to express that previously reported post-treating methods were limited to depositing a physically stacked passivation layer on top of the 3D perovskite layer, but the bulk 3D perovskite couldn't be improved by these methods. The sayings in the last version seem to exclude the work that adding 2D into the perovskite bulk. Adding additives to the bulk of 3D perovskite and post-treatment by surface passivation are two important ways to improve the quality of perovskite films. The scientific issues in these two methods are quite different. This work focuses on developing a novel surface passivation method that concurrently passivates the surface defects and induces secondary crystal growth in the bulk 3D perovskite. Therefore, we restricted the claim in the abstract to "previous reports that studying perovskite post-treatment". The following changes were made in the revised manuscript.

Changes in the manuscript: Constructing 2D/3D perovskite heterojunctions is effective for the surface passivation of perovskite solar cells (PSCs). However, previous reports that studying perovskite post-treatment only physically deposits 2D perovskite on the 3D perovskite, and the bulk 3D perovskite remains defective. Herein, we propose Cl₂-dissolved chloroform as a multifunctional solvent for concurrently constructing 2D/3D perovskite heterojunction and inducing the secondary growth of the bulk grains.

Comment 2. The XRD peak shift from 14.07 to 14.09 is very small? Is it within the resolution of the equipment?

Response: Thanks for the reviewer's comment. We consulted SmartLab's engineer and confirmed that the resolution of the X-ray diffractometer was 0.01 degree. Therefore, the small shift from 14.07 to 14.09 was resolvable and meaningful. It was

reported that a similar small shift of XRD peaks (about 0.01 to 0.02 degree) could be induced by adding different amounts of MACl into the precursor solution of 3D perovskites (Ye F, et al. *Adv Mater*, **2021**, 33 2007126).

Comment 3. It seems like the device performance is enhanced due to the synergetic effect of Cl^- diffusion and n-HABr passivation. What's about using n-HACl passivation? Would it have similar effect?

Response: Thanks for this constructive suggestion. We compared the passivation effect of n-HACl dissolved in pure chloroform without Cl_2 , n-HACl dissolved in Cl_2 -dissolved chloroform, and n-HABr dissolved in Cl_2 -dissolved chloroform. As shown in **Supplementary Fig. 24**, n-HACl dissolved in fresh chloroform to a certain extent gave some performance improvement in V_{oc} and PCE, which was most likely due to the synergetic effect of Cl^- diffusion and n-HACl passivation. However, n-HACl and n-HABr dissolved in Cl_2 -dissolved chloroform exhibited better average V_{oc} and PCE than n-HACl dissolved in fresh chloroform, indicating that the dissolved Cl_2 played an important role. This was most likely due to Cl_2 -induced secondary crystal growth that further reduced the defects in the bulk of the 3D perovskite.

Supplementary Fig. 24 | Photovoltaic parameters of the PVSF, HACl/CF-treated, HACl/Cl₂-CF-treated, and HABr/Cl₂-CF-treated devices.

Changes in the manuscript:

To further investigate the effect of Cl₂, we compared the performance of the devices based on perovskite post-treatment by n-HACl dissolved in pure chloroform without Cl₂, n-HACl dissolved in Cl₂-dissolved chloroform, and n-HABr dissolved in Cl₂-dissolved chloroform. As shown in **Supplementary Fig. 24**, n-HACl and n-HABr dissolved in Cl₂-dissolved chloroform exhibited better average V_{OC} and PCE than n-HACl dissolved in fresh chloroform, indicating that the dissolved Cl₂ played an important role. This was most likely due to Cl₂-induced secondary crystal growth that further reduced the defects in the bulk of the 3D perovskite.

Comment 4. There is no Cl in the control perovskite composition, why did the authors observe up to 0.25% Cl content using XPS? How did the author characterize the buried interface without damaging the perovskite film? This point needs to be clearly mentioned in the manuscript.

Response: Thanks for the reviewer's comments. The control perovskite films were deposited by the two-step spin-coating method. In the second step, MACl was added to optimize the crystallization. The details can be found in the experimental section. It was reported that MACl was volatile and most MACl eventually escaped from the perovskite (Kim, M. *et al. Joule* **2019**, 3 2179-2192), but a small amount of Cl⁻ could still be detected (Min, H. *et al. Nature* **2021**, 598, 444-450).

To expose and characterize the buried interface without damaging of the perovskite film, the previously reported technique was used (Yang, X. *et al. Adv. Mater.* **2021**, 33, 2006435). Briefly, we first fabricated perovskite samples with the structure of ITO/PTAA/perovskite/Ag. And then the samples were immersed in chlorobenzene (CB) for 20 min in the N₂-filled glovebox. After the PTAA layer was dissolved by CB, the perovskite/Ag film detached from the substrate and floated on the surface of CB. Finally, the lift-off perovskite film was fixed to a base with its bottom side.

Changes in the manuscript:

Change 1: The corresponding energy dispersive spectroscopy (EDS) results in **Fig. 2b** revealed that the PVSK-Cl₂-CF exhibited the highest Cl content of 2.64% at the bottom interface (vs. 0.28% for PVSK, 0.25% for PVSK-CF). Note that MA₂Cl was used to adjust the crystallization of perovskite, so a small amount of Cl was detected in the PVSK sample.

Changes 2:

Film deposition and device fabrication.

The method to get the buried interface of perovskite film without damage: To expose and characterize the buried interface without damaging the perovskite film, the previously reported technique was used.⁵⁶ Briefly, perovskite samples with the structure of ITO/PTAA/perovskite/Ag were first fabricated. And then the samples were immersed in CB for 20 min in the N₂-filled glovebox. After the PTAA layer was dissolved by CB, the perovskite/Ag film detached from the substrate and floated on the surface of CB. Finally, the lift-off perovskite film was fixed to a base with its bottom side.

Comment 5. How did the author measure the tDOS in the device? This need to be explained in the manuscript.

Response: Thanks for the reviewer's suggestion. The tDOS ($N_t(E_\omega)$) of the device was determined by measuring the electrochemical impedance spectroscopy (EIS) and the Mott-Schottky curve in the dark (Wang, M. *et al. Adv. Funct. Mater.* **2022**, 32, 2108567). The details were added in the Methods of the revised manuscript.

Changes in the manuscript:

Methods

Calculation of $N_t(E_\omega)$. The tDOS ($N_t(E_\omega)$) of the device was determined by measuring the impedance spectroscopy (EIS) and the Mott-Schottky curves in the dark using the previously reported method.⁵⁷ The N_t can be estimated by equation (4):

$$N_t(E_\omega) = -\frac{\omega}{K_B T} \times \frac{V_{bi}}{eW} \times \frac{dC}{d\omega} \quad (4)$$

where ω was the angular frequency, K_B was the Boltzmann constant, T was the temperature, V_{bi} was the built-in electric field, e was the electron charge, W was the depletion width and C was the capacitance. The independent variable of energy E_ω can be determined by equation (5):

$$E_\omega = K_B T \times \ln \frac{2\beta_p N_V}{\omega} \quad (5)$$

where β_p was the capture coefficient of hole, N_V was the effective density of states in the valence band. The depletion layer width W can be calculated by equation (6):

$$W = \sqrt{\frac{2\varepsilon_s}{q} \times \frac{N_A + N_D}{N_A \times N_D} \times V_{bi}} \quad (6)$$

where ε_s was the dielectric constant of perovskite active layer, N_A and N_D were the doping concentrations of the hole-transporting layer and the electron-transporting layer respectively. V_{bi} can be determined by measuring the Mott-Schottky curve and calculated by equation (7):

$$\frac{1}{C^2} = \frac{2}{n^2} \times \frac{1}{q\varepsilon_s} \times \frac{N_A + N_D}{N_A \times N_D} (V - V_{bi}) \quad (7)$$

where n was a constant of proportionality. Then the formula for k was expressed as:

$$k = \frac{2}{n^2} \times \frac{1}{q\varepsilon_s} \times \frac{N_A + N_D}{N_A \times N_D} \quad (8)$$

Therefore, equation (6) was simplified to (9).

$$W = \varepsilon_s \times \sqrt{-kn^2 \times V_{bi}} \quad (9)$$

In addition, the geometric capacitance C_g can be obtained from the high-frequency region of the EIS data by equation (10):

$$C_g = \frac{n\varepsilon_s}{d} \quad (10)$$

where d was the thickness of the perovskite layer, then equation (3) can again be expressed as:

$$W = C_g \times d \times \sqrt{-kV_{bi}} \quad (11)$$

Therefore, β_p , N_v , ε_s , C_g , d , k , V_{bi} , and $\frac{dC}{d\omega}$ of related materials should be known to calculate N_t . The value of 10^{-8} cm³/s for β_p and the value of 2.524×10^{-19} /cm³ for N_v was used in the calculation.⁵⁷

Comment 6. In terms of stability, the authors only demonstrate enhanced operation stability. The authors need to demonstrate ambient stability and thermal stability as well.

Response: Thanks for this important comment. To confirm the effect of HABr/Cl₂-CF treatment on device stability, we further conducted additional experiments about ambient stability and thermal stability.

Changes in the manuscript:

Change 1: We further tracked the ambient stability of the devices (**Supplementary Fig. 28**). After 1920 hours of aging in the air with a relative humidity of 10% and a temperature of ~25 °C, the unencapsulated device based on PVSK-HABr/Cl₂-CF maintained 91% of its original PCE while the device based on PVSK and PVSK-HABr/CF maintained 78% and 82% respectively. **Supplementary Fig. 29** and **Supplementary Fig. 30** showed that HABr/Cl₂-CF treatment improved the thermal stability with the devices based on PVSK-HABr/Cl₂-CF remained over 92% of the initial PCE after heating for 400 hours at 55 °C in a nitrogen environment and remained over 81% of the initial PCE after heating for 106 hours at 85 °C in a nitrogen environment.

Change 2:

Film deposition and device fabrication. For the thermal stability test, a polymer (PBDB-T)-doped Spiro-OMeTAD hole-transporting layer was employed to improve the thermal stability of the devices.

Changes in the revised SI:

Supplementary Fig. 28 | Ambient stability of the PVSK, PVSK-HABr/CF, and PVSK-HABr/Cl₂-CF based devices. The devices were stored in the air with a relative humidity of ~10% and a temperature of ~25 °C. All the error bars represent the standard deviation for 8 devices.

Supplementary Fig. 29 | Thermal stability (55 °C) of the PVSK, PVSK-HABr/CF, and PVSK-HABr/Cl₂-CF-based devices. All the error bars represent the standard deviation for 8 devices.

Supplementary Fig. 30 | Thermal stability (85 °C) of the PVSK, PVSK-HABr/CF, and PVSK-HABr/Cl₂-CF-based devices. All the error bars represent the standard deviation for 5 devices.

Reviewer #2

This paper reports that chloroform is oxidized to form Cl_2 , and the Cl_2 formed in this way is treated on the surface of perovskite to penetrate not only the surface but also the bulk, effectively passivating defects, and growing crystal grains. The approach is interesting and the influence of Cl_2 is clear from the experimental results, but there are critical concerns about reaching conclusions. I believe that only when these concerns are addressed can we decide whether to publish the paper.

General response: We sincerely thank the reviewer for making many valuable comments for further improving the quality of our manuscript. We have comprehensively revised the manuscript according to the reviewer's suggestions.

Comment 1. [p6. Line 127-135] In the manuscript, the authors described that the Cl incorporation shifts the peak of FAPbI_3 from 14.07 to 14.09 but Supplementary Fig.9a shows the opposite shift from 14.07 to 14.04. The peak of PbI_2 is too. Data in Supplementary Fig 9a reveals that Cl_2 -CF treatment increases the lattice parameter of FAPbI_3 which is in contrast to the substitution of smaller Cl ions into I sites in FAPbI_3 . This is critical because it is the opposite of the main insist of this paper.

Response: Thanks for the reviewer's important comment. We have re-checked the related XRD data and repeated the XRD experiment. We found that we made a mistake by using the wrong data in **Supplementary Fig. 9** in the previous version. The updated XRD results showed that Cl_2 -CF treatment can induce a shift of the XRD peak of α - FAPbI_3 from 14.00 ° to 14.05 ° and the shift of the PbI_2 peak from 12.70 ° to 12.75 °. This was due to the substitution of smaller Cl^- ions into I sites, which led to the lattice contraction of the perovskite crystal.

Supplementary Fig. 9 | XRD patterns of PVSK, PVSK-CF, and PVSK-Cl₂-CF. (a) 5 ° to 40 °, (b) 12 ° to 15 °, (c) 12.2 ° to 13.0 °.

Changes in the manuscript: We performed X-ray diffraction (XRD) measurements to study the effect of Cl₂-CF on the crystal structure of perovskite. After the Cl₂-CF treatment, compared to PVSK, PVSK-Cl₂-CF exhibited higher XRD peak intensity, indicating improved crystallization (Supplementary Fig. 9a). Due to the ionic exchange of I⁻ and Cl⁻, XRD patterns of α-FAPbI₃ located at 14.00 ° shifted to 14.05 ° in PVSK-Cl₂-CF, indicating the formation of α-FAPbI_(3-x)Cl_x (Supplementary Fig. 9b). Similarly, the XRD peak of the residual PbI₂ revealed a peak shift from 12.70 ° to 12.75 °, indicating the formation of PbI_(2-x)Cl_x (Supplementary Fig. 9c).

Comment 2. The authors insist that Cl₂ plays an important role in the passivation and grain growth of FAPbI₃ film which is confirmed by analysis in Figure 1. In addition, the authors insist that in the HABr/Cl₂-CF treatment, Cl₂ plays the same role as the Cl₂-CF treatment. But as shown in Supplementary Fig 6b, when HABr is dissolved in Cl₂-CF, Cl₂ produces Br₂. If the state of Cl₂ in the HABr/Cl₂-CF solution is different from the Cl₂-CF solution, the effect of Cl₂ could not be interpreted by only diffusion into the bulk. There could be an effect of Br₂.

Response: Thanks very much for the important comment. We agree with the reviewer that when HABr was dissolved in Cl₂-CF, Cl₂ produced Br₂ and it was reasonable to speculate that Br₂ may have additional effects.

We first studied the state of Cl₂ in the HABr/Cl₂-CF solution. As shown in Supplementary Fig. 16, when placed the wet starch potassium iodide test paper

above the HABr/Cl₂-CF solution, it turned blue immediately. This result indicated that although partial Cl₂ took part in oxidizing the Br⁻ ions of HABr, there was residual Cl₂ in the solution.

Therefore, compared to the Cl₂-CF solvent, Cl₂ in the HABr/Cl₂-CF solution can be divided into two parts. One part took part in oxidizing Br⁻ ions to Br₂ and leaving Cl⁻ ions in the solution. The other part was the residual Cl₂ in Cl₂-CF solvent, which would penetrate the depth of perovskite films to trigger the redox reaction with perovskite, inducing the Cl-doping and secondary growth of perovskite crystal grains.

During the post-treatment experiment using HABr/Cl₂-CF solution, we observed a light-yellow color for the as-treated perovskite films (**Supplementary Fig. 17**), which can be attributed to the existence of Br₂ on the perovskite surface. After annealing at 100 °C for about 10 min, the light-yellow color disappeared, which could be attributed to the volatilization of Br₂ species.

To investigate the effect of Br₂ on the passivation effect of the HABr/Cl₂-CF solution, we also added different mass ratios of Br₂ into the solution of HABr dissolved in fresh CF. Note that no Br₂ can be produced in HABr/CF solution due to there being no Cl₂ in the fresh CF. So the effect of the added Br₂ can be studied. The statistics of the devices were shown in **Supplementary Fig. 25**. It demonstrated that Br₂ did not improve the V_{OC} and FF of the devices. Although Br₂ can also oxidize I⁻, the effect of Br₂ on the device performance was very different from that of Cl₂. This can be attributed to the weaker oxidability and lower penetration ability due to the higher boiling point of Br₂. As indicated by the color change after spin-coating, most Br₂ may remain on the surface of perovskite film and give litter effects.

Supplementary Fig. 16 | Comparison of the state of Cl₂ in the Cl₂-CF solvent and the HABr/Cl₂-CF solution by testing the vapor with wet starch potassium iodide test paper. The blue color proved the existence of Cl₂ in HABr/Cl₂-CF solution although partial Cl₂ was consumed to oxidize the Br⁻ ions to Br₂.

Supplementary Fig. 17 | Photographs of (a) PVSK, (b) PVSK-HABr/CF, (c) PVSK-HABr/Cl₂-CF, (d) PVSK-HABr/CF Annealed, and (e) PVSK-HABr/Cl₂-CF Annealed.

Supplementary Fig. 25 | Photovoltaic parameters of the devices post-treated by HABr/CF containing different ratios of Br₂ as additives. (a) V_{OC}, (b) J_{SC}, (c) fill factor and (d) efficiency.

Changes in the manuscript:

Change 1: To investigate whether the formed Br₂ in the HABr/Cl₂-CF solution affected the device performance, we started by studying the state change of Cl₂ in the HABr/Cl₂-CF solution (Supplementary Fig. 16). When placing the wet starch potassium iodide test paper above the HABr/Cl₂-CF solution, it turned blue immediately. This result indicated that although partial Cl₂ took part in oxidizing the Br⁻ ions of HABr to be Br₂, there was residual Cl₂ in the solution. Therefore, compared to the Cl₂-CF solvent, Cl₂ in the HABr/Cl₂-CF solution can be divided into two parts. One part took part in oxidizing Br⁻ ions to Br₂ and left Cl⁻ ions in the solution. The other part was the residual Cl₂ in Cl₂-CF solvent, which would penetrate the depth of perovskite films to trigger the redox reaction with perovskite, inducing the Cl-doping and secondary growth of perovskite crystal grains. During the post-treatment experiment using HABr/Cl₂-CF solution, we observed a light-yellow color for the as-treated perovskite films (Supplementary Fig. 17), which can be attributed to the existence of Br₂ on the perovskite surface. After annealing at 100 °C for about

10 min, the light-yellow color disappeared, which could be attributed to the volatilization of Br₂ species.

Change 2: To investigate whether the formed Br₂ in the HABr/Cl₂-CF solution affected the passivation effect, we added different mass ratios of Br₂ into the solution of HABr dissolved in CF. Note that no Br₂ can be produced in the PVSK and HABr/CF due to there being no Cl₂ in the CF. So the effect of the added Br₂ can be studied. The statistics of the devices were shown in **Supplementary Fig. 25**. It demonstrated that Br₂ did not improve the V_{OC} and FF of the device. Although Br₂ can also oxidize I⁻, the effect of Br₂ on the device performance was very different from that of Cl₂. This can be attributed to the weaker oxidability and lower penetration ability due to the higher boiling point of Br₂. As indicated by the color change after spin-coating, most Br₂ may remain on the surface of perovskite film and give litter effects.

Comment 3. Since this work fabricates FAPbI₃ by sequential method, the control film includes a lot of PbI₂. This is unusual. In most recent papers related to FAPbI₃, the thin film surface in SEM images show clear without PbI₂ which is in contrast to this work in Supplementary Fig. 11. In this paper, it is difficult to exclude the effect of the removal of PbI₂ during post-treatment, making the argument of the paper unclear. Therefore, experiments on complete FAPbI₃ thin films without PbI₂ are essential to clarify the conclusion of this paper. Complete FAPbI₃ thin films can be produced by either the two-step method [DOI: 10.1126/science.aaa9272, DOI: 10.1126/sciadv.abe3326] or by one-step method[<https://doi.org/10.1038/nature14133>].

Response: Thanks for the reviewer's constructive comment. We agree with the reviewer that PbI₂-free FAPbI₃-based perovskite films can be prepared by both the two-step method (*Science* **2015**, 348, 1234; *Sci. Adv.* **2021**, 7, eabe3326) and the one-step method (*Nature* **2015**, 517, 476). In the sequential two-step method adopted in this work, You *et. al.* reported that both PbI₂-free perovskite film and perovskite with different amounts of PbI₂ can be prepared by simply adjusting the spinning speed of

the first and second steps (*Adv. Mater.* **2017**, 29, 1703852). They found that moderate residual of PbI₂ can deliver the best device performance with improved stability and reduced hysteresis. Starting from the similar PbI₂-contained FAPbI₃-based perovskite film prepared by the two-step method, the same group reported a high certified PCE of 25.6% (*Science* **2022**, 377, 531-534). The passivation effect of a small amount of residual PbI₂ was also reported in the perovskite prepared by the one-step method (*Sci. Adv.* **2016**, 2, e1501170; *Adv. Funct. Mater.* **2023**, 2215171; *Angew. Chem. Int. Ed.* **2022**, 61, e202115663).

In our group, the composition and depositing conditions of the control perovskite were optimized in the previously published works (*Nat. Commun.* **2022**, 13, 4891; *Adv. Energy Mater.* **2023**, 13, 2204362; *J. Mater. Chem. A.* **2021**, 9, 20807). So we continued to use this optimized control perovskite with a small amount of residual PbI₂ in this work. We agree with the reviewer that PbI₂-contained perovskite may to a certain extent complicate studying the effect of Cl₂. However, we think that starting from the optimized control with higher device performance will make the work more attractive. Moreover, due to the passivation effect of the residual PbI₂, it is reasonable to argue that the removal of PbI₂ during post-treatment will most likely gave a deleterious effect. So this aspect will perhaps not weaken the arguments regarding the role of Cl₂ in the improvement of device performance in this work.

According to the reviewer's suggestion, we also studied the effect of HABr/Cl₂-CF treatment on the PbI₂-free (or PbI₂-less) perovskite. As shown in the SEM images in **Supplementary Fig. 12**, no distinct secondary growth of crystal grains can be observed. This result indicated that during the treatment by the HABr/Cl₂-CF solution, Cl₂ was most likely first reacted with the residual PbI₂ and this reaction triggered the subsequent secondary grain growth, leading to high-quality perovskite films with larger grains.

Supplementary Fig. 12 | SEM images of perovskite films without and with HABr/Cl₂-CF treatment. The perovskite films were fabricated by the one-step spin-coating method and showed almost a PbI₂-free surface.

Changes in the manuscript: The roughness of the PVSK-HABr/Cl₂-CF is largely reduced to 26.2 nm, which was comparable with that of 25.7 nm for the PVSK-HABr/CF and much less than that of 32.9 nm for the PVSK (**Supplementary Fig. 11 g-i**). The effect of HABr/Cl₂-CF treatment on the PbI₂ free (or PbI₂ less) was also studied. As shown in the SEM images in **Supplementary Fig. 12**, no distinct secondary growth of crystal grains can be observed. This result indicated that during the treatment by the HABr/Cl₂-CF solution, Cl₂ was most likely first reacted with the residual PbI₂ and this reaction triggered the subsequent secondary grain growth, leading to high-quality perovskite films with larger grains.

Reviewer #3

In this manuscript, authors report a new interface passivation strategy using solvent engineering method. Through Cl₂-dissolved CF, authors provide the effective construction of the 2D/3D perovskite heterojunction. Also, this method enables high efficiency of over 24% with enhanced operational stability. Additionally, authors have performed appropriate analyzes to support their claims for reduced defect with this new passivation strategy. This new passivation strategy is obviously novel and different from the papers published so far. Thus, I recommend that this manuscript is published in Nature communications. Some aspects of the work may deserve the

authors' further attention and then lead to revision:

General response: We thank the reviewer for reviewing the manuscript and giving valuable comments. We have carefully revised the manuscript following the reviewer's suggestions.

Comment 1. Clearly, this strategy will give different passivation effects at grain and grain boundary. Therefore, 2D PL mapping analysis could further enhance the quality of this manuscript.

Response: Thanks for this constructive comment. We performed 2D PL mapping measurements and the images were shown in **Supplementary Fig. 18**. The results showed that compared with the PVSK and PVSK-HABr/CF samples, the PVSK-HABr/Cl₂-CF sample showed overall higher PL intensity as well as larger perovskite grains, indicating better passivation effect and secondary grain growth. Moreover, there were a lot of dark grain boundaries for the PVSK and PVSK-HABr/CF samples while most of them disappeared in the PVSK-HABr/Cl₂-CF sample. These results indicated that the HABr/Cl₂-CF treatment led to a superior passivation effect at grain boundaries.

Supplementary Fig. 18 | 2D PL mapping images of (a) PVSK, (b) PVSK-HABr/CF, and (c) PVSK-HABr/Cl₂-CF.

Changes in the manuscript:

Change 1: The 2D PL mapping images shown in **Supplementary Fig. 19** demonstrated that PVSK-HABr/Cl₂-CF showed overall higher PL intensity as well as larger perovskite grains, indicating better passivation effect and second grain growth.

Moreover, there were a lot of dark grain boundaries for the PVSK and PVSK-HABr/CF samples while most of them disappeared in the PVSK-HABr/Cl₂-CF sample. These results indicated that the HABr/Cl₂-CF treatment led to a superior passivation effect at grain boundaries.

Change 2: Characterization. PL mapping images were acquired using a laser confocal microscope (Leica TCS SP8) excited at a 488 nm pulse.

Comment 2. The grain size through SEM appears larger, but the FWHM of XRD does not seem to change significantly. Please compare the FWHM of the XRD to make sure the grain size of bulk is increased, not just the surface, and calculate the grain size from FWHM.

Response: Thanks for the reviewer's important suggestion. We compared the grain size of the bulk using the FWHM of the XRD. The relationship between the grain size and the FWHM of the XRD peak can be expressed by the Scherrer equation:

$$D = \frac{K\lambda}{FWHM \cdot \cos\theta} \quad (9)$$

where D is the mean crystallite size, K is the Scherrer constant, λ is the wavelength of the incident X-ray, FWHM is the full-width at half-maximum of the XRD peak (with instrumental broadening removed), and θ is the Bragg angle.

Table R1 2θ , FWHM, and D of the PVSK, PVSK-HABr/CF, and PVSK-HABr/Cl₂-CF.

	PVSK	PVSK-HABr/CF	PVSK-HABr/Cl ₂ -CF
2θ (degree)	14.01	14.01	14.06
FWHM (degree)	0.091	0.092	0.087
D (nm)	87.02	87.02	91.02

The results were shown in **Table R1**. It can be seen that the FWHM of the PVSK, PVSK-HABr/CF, and PVSK-HABr/Cl₂-CF were 0.091, 0.092, and 0.087 degree, respectively. So the calculated average grain size was 87.02, 87.02, and 91.02 nm, respectively. Although the evolution of the grain size agreed with that obtained from the SEM images, the absolute values showed a major mismatch. Note that the Scherrer equation for determining the crystallite size works well in the sub-micrometer range. (M. Leoni, in *International Tables for Crystallography* (John Wiley & Sons, Ltd., 2019), p. 524.). The authors of the book recommended not to use the Scherrer analysis for microstructural analysis unless there was solid evidence proving the absence of structural defects in the sample. As the grain size of the perovskite film reached several microns (about 3.5 μm for the PVSK-HABr/Cl₂-CF), the average *D* obtained by the Scherrer equation may be not accurate.

As shown in **Supplementary Fig. 10**, the SEM image of the bottom side of the perovskite film also showed increased grain size for the PVSK-HABr/Cl₂-CF, which indicated that the grain growth was not only occurred at the surface of the film but in the bulk of the whole film.

Comment 3. This manuscript employs HABr to form 2D/3D structures. Therefore, the papers introducing HABr for the first time should be cited (DOI: 10.1039/C9EE00751B) and explained with its advantages.

Response: Thanks very much for this important comment. Yoo *et al.* first utilized HABr as the 2D perovskite precursor to construct the 2D/3D structure (*Energy Environ. Sci.* **2019**, 12, 2192). They found that HABr can be selectively dissolved in chloroform (CF), which was non-solvent for perovskite films. Therefore, treating the 3D perovskite films with HABr/CF could construct a 2D perovskite layer without damage that was commonly observed in the post-treatment by using alkyl ammonium salt dissolved in IPA. Therefore, this strategy can help fabricate high-performance perovskite solar cells with improved stability.

Changes in the manuscript:

Change 1: The most widely used method for constructing 2D/3D heterojunctions is spin-coating bulky alkyl ammonium halides onto the 3D perovskite, followed by a thermal annealing process. For example, Yoo *et al.* first utilized hexylammonium hydrobromide (HABr) as the 2D perovskite precursor to construct the 2D/3D structure.³¹ They found that HABr can be selectively dissolved in chloroform (CF), which was a non-solvent for perovskite films. Therefore, treating the 3D perovskite films with HABr/CF could construct a 2D perovskite layer without damage that was commonly observed in the post-treatment by using alkyl ammonium salt dissolved in isopropanol (IPA). Therefore, this strategy can help fabricate high-performance perovskite solar cells with improved stability. However, the resultant 2D perovskite capping layer in previous reports was most likely physically stacked onto the 3D perovskite, and only weak interfacial interactions existed between the 2D and 3D perovskite layers.

Change 2: Herein, we report adopting chlorine-dissolved chloroform (Cl₂-CF) as a multifunctional solvent for selectively dissolving HABr to construct 2D/3D perovskite heterojunction, as well as to induce secondary growth of perovskite grains and defect passivation through the redox reaction between Cl₂ and I.

REVIEWERS' COMMENTS

Reviewer #1 (Remarks to the Author):

The authors have sufficiently addresses my comments and concern. In my opinion, the quality of the manuscript has been significantly improved, and the results are significant enough to be published in Nature Com now. I recommend the publication of this in Nature Com as is.

Reviewer #2 (Remarks to the Author):

The authors seem to have diligently conducted additional experiments to address the concerns raised. Based on the various results presented by the authors, it appears that the addition of Cl₂ has a positive effect on the perovskite film. The biggest concern, which was the evidence of Cl ions substituting I sites, has been confirmed by new XRD results and a blue shift in the EQE spectrum. Other concerns have also been effectively addressed through additional experiments. Although there have been many reports on the effects of Cl ions on perovskite thin films prior to this study, the significance of this paper lies in the approach of demonstrating how to achieve the effect through surface treatment. Therefore, I recommend the publication of this paper.

REVIEWERS' COMMENTS

Reviewer #1 (Remarks to the Author):

The authors have sufficiently addresses my comments and concern. In my opinion, the quality of the manuscript has been significantly improved, and the results are significant enough to be published in Nature Com now. I recommend the publication of this in Nature Com as is.

Response: We thank the reviewer for reviewing the work and giving constructive comments.

Reviewer #2 (Remarks to the Author):

The authors seem to have diligently conducted additional experiments to address the concerns raised. Based on the various results presented by the authors, it appears that the addition of Cl₂ has a positive effect on the perovskite film. The biggest concern, which was the evidence of Cl ions substituting I sites, has been confirmed by new XRD results and a blue shift in the EQE spectrum. Other concerns have also been effectively addressed through additional experiments. Although there have been many reports on the effects of Cl ions on perovskite thin films prior to this study, the significance of this paper lies in the approach of demonstrating how to achieve the effect through surface treatment. Therefore, I recommend the publication of this paper.

Response: We thank the reviewer for reviewing the work and giving valuable comments.